# Metabolic Profiling for Unveiling Mechanisms of Kushenol F against Imiquimod-Induced Psoriasis with UHPLC/MS Analysis

**DOI:** 10.3390/molecules29112410

**Published:** 2024-05-21

**Authors:** Xingxin Yang, Jiaoli Cheng, Xunqing Yin, Ting Ao, Xudong He, Yaqin Yang, Yuping Lin, Zhen Chen

**Affiliations:** 1College of Pharmaceutical Science, Yunnan University of Chinese Medicine, Kunming 650500, China; yxx78945@163.com (X.Y.); yinxunqing1018@163.com (X.Y.); ace153790@163.com (T.A.); hexudong@ynutcm.edu.cn (X.H.); 2Science and Technology Achievement Incubation Center, Kunming Medical University, Kunming 650500, China; jiaoli_cheng980217@163.com; 3School of Pharmaceutical Sciences, Zhejiang Chinese Medical University, Hangzhou 311403, China

**Keywords:** inflammation, Kushenol F, psoriasis, metabolomics, *Sophora flavescens*

## Abstract

Psoriasis is a common chronic immune-mediated inflammatory skin disorder. *Sophora flavescens* Alt. (*S. flavescens*) has been widely acknowledged in the prevention and treatment of psoriasis. Kushenol F (KSCF) is a natural isopentenyl flavonoid extracted from the root of *S. flavescens*. We aimed to investigate the effect and mechanism of KSCF on imiquimod (IMQ)-induced psoriasis-like skin lesions in mice. A mouse model of psoriasis was induced with 5% IMQ for 5 days, and the mice were given KSCF dermally for 5 days. Changes in skin morphology, the psoriasis area, the severity index (PASI), and inflammatory factors of psoriasis-like skin lesions were evaluated. Metabolites in the psoriasis-like skin lesions were analyzed with ultra-high-performance liquid chromatography/mass spectrometry followed by a multivariate statistical analysis to identify the differential metabolites and metabolic pathway. The results of the present study confirmed that KSCF significantly reduced PASI scores, epidermal thickening, and epidermal cell proliferation and differentiation. KSCF also reduced the levels of interleukin (IL)-1β, IL-6, IL-8, IL-17A, IL-22, IL-23, and tumor necrosis factor (TNF)-α in the injured skin tissues while increasing IL-10 content. KSCF significantly regulated metabolites in the skin samples, and a total of 161 significant metabolites were identified. These differential metabolites involved sphingolipid and linoleic acid metabolism and steroid hormone biosynthesis. Collectively, KSCF inhibited the inflammatory response to prevent IMQ-induced psoriasis-like skin lesions in mice by call-backing the levels of 161 endogenous metabolites and affecting their related metabolic pathways. KSCF has the potential to be developed as a topical drug for treating psoriasis symptoms.

## 1. Introduction

Psoriasis is a chronic and systemic immune-mediated disease. The underlying pathomechanisms involve an interplay between the innate and adaptive immune systems [1,2]. Recently, psoriasis presents many challenges, including high prevalence, chronicity, and associated comorbidity [3]. As a representative inflammatory skin disease occupied by large surface involvement, the occurrence and development of psoriasis is reported to be related to the disorder of many inflammation markers, such as IL-17A, IL-23, and TNF-α. Biologics targeting IL-17A, IL-23, and TNF-α have been developed and approved for the treatment of psoriasis [4]. The progression of psoriasis has also been confirmed to be related to immunity [5,6,7].

Common treatments for psoriasis include topical agents, systemic therapy, and ultraviolet light therapy. Topical agents mainly include calcipotriol (CVD), corticosteroids, and salicylic acid. Methotrexate, cyclosporine, and acitretin are commonly used as systemic therapy for psoriasis [8,9]. Although the therapeutic effects of these drugs have been widely recognized, adverse reactions still largely limit their clinical utilization [10]. Hence, complementary and alternative medicine may offer options for some patients.

*Sophora flavescens* Alt. (*S. flavescens*) is widely used in the clinic, can clear dampness-heat, kill insects, and dispel wind evil, which originates from the teachings of traditional Chinese medicine [11]. *S. flavescens* has been clinically used in numerous countries to treat skin diseases [12,13]. Kushenol F (KSCF) is a natural isopentenyl flavonoid extracted from the root of *S. flavescens*. It has been reported to have anti-inflammatory and anti-bacterial properties [14,15,16]. A previous study demonstrated that KSCF oral treatment could be an important therapeutic for treating atopic dermatitis. Although various studies have reported the biological effects of KSCF, the effect and mechanism of KSCF on psoriasis are unclear.

Metabolomics can reveal the specific changes in physiological and biochemical states related to phenotypes in biological systems. The application of skin metabonomics is suitable for revealing the potential functional changes in a variety of metabolic pathways and signal networks in psoriasis-like organisms, which has become a significant approach to evaluate the efficacy and mechanism of drugs [17]. The present study investigated the effects of KSCF on imiquimod (IMQ)-induced psoriasis-like skin lesions in mice using conventional pharmacology and metabolomics. The potential biomarkers and complex mechanisms of KSCF involved in psoriasis treatment are also discussed. The results demonstrated that KSCF inhibited the inflammatory response to prevent IMQ-induced psoriasis-like skin lesions in mice by call-backing the levels of 161 endogenous metabolites and affecting their related metabolic pathways. Thus, KSCF has the potential to be developed as a topical drug for treating psoriasis symptoms.

## 2. Results and Discussion

### 2.1. KSCF Mitigated the Psoriatic Dermatitis Induced by IMQ

Psoriasis is a skin disease with characteristics of well-demarcated, erythematous, raised lesions with silvery-white dry scales. As depicted in Figure 1, after 5 days of IMQ treatment, erythema and silvery-white dry scales were observed on the dorsal skin, and the PASI was significantly increased (*p* < 0.001). However, after treatment with CVD and KSCF, the thickness and size of the scales on the dorsal skin were decreased. Erythema and PASI were also significantly decreased (*p* < 0.001). These results indicate that KSCF effectively relieved IMQ-induced psoriasis symptoms.

### 2.2. Effect of KSCF on Inflammatory Reactions in Psoriasis-like Mice

Psoriasis is a chronic inflammatory skin disease characterized by lymphocyte and neutrophil infiltration and the excessive keratinization and proliferation of keratinocytes [18,19]. The agents capable of attenuating keratinocyte hyperproliferation and excessive inflammatory responses are considered to be potentially useful for psoriasis treatment [20,21]. IMQ-induced psoriasis is usually accompanied by cellular inflammatory factor disorders in the dorsal skin. As shown in Figure 2A, IMQ induced massive inflammatory cell infiltration in the dorsal skin, which was reduced by CVD and KSCF treatment. As shown in Figure 2B, after 5 days of IMQ treatment, the levels of IL-1β, IL-6, IL-8, IL-17A, IL-22, IL-23, and TNF-α significantly increased in the dorsal skin (*p* < 0.05), whereas IL-10 levels significantly decreased (*p* < 0.0001). After treatment with CVD and KSCF, the levels of IL-1β, IL-6, IL-8, IL-17A, IL-22, IL-23, and TNF-α significantly decreased (*p* < 0.05) and IL-10 levels significantly increased (*p* < 0.0001). These results indicated that KSCF might stimulate the secretion of anti-inflammatory cytokines while inhibiting the release of pro-inflammatory cytokines, thereby alleviating IMQ-induced psoriasis.

### 2.3. Effect of KSCF on Metabolites in the Injured Skin of Psoriasis-like Mice

The data of skin samples were collected with ultra-high-performance liquid chromatography/mass spectrometry (UHPLC/MS), and the metabolic profiles of each group were obtained. As depicted in Appendix A, the peak number and intensity significantly varied in the total ion current profiles of the control, model, CVD, and KSCF groups, indicating significant differences among the metabolic state of each group and endogenous metabolites. Thus, the endogenous metabolites of skin changed significantly after IMQ induction and administration. In this study, PCA and OPLS-DA were used to analyze metabolome differences in the dorsal skin between the control, model, and KSCF groups using an unsupervised statistical method. The metabolic state of the control and model groups was different (Figure 3 and Figure 4A). The KSCF group was far from the model group. The results demonstrated that KSCF effectively reversed the pathological changes triggered by IMQ treatment in psoriasis-like mice.

Two hundred iteration permutation tests were performed on OPLS-DA in the positive and negative ion modes to further illustrate the reliability of the OPLS-DA model. The Q^2^ values were less than 0.05, further confirming the accuracy of the multivariate statistical analysis (Figure 4B).

### 2.4. Identification of Differential Metabolites

Changes in the metabolites can reflect the physiological and pathological states of the human body and might serve as an objective index to determine the efficacy and mechanism of drugs during disease intervention [22]. The complete metabolic information was collected, and the mechanism of KSCF against IMQ-induced psoriasis, which involved the regulation of endogenous metabolites, was revealed using the UHPLC/MS technique. S-plot diagrams based on OPLS-DA analysis were plotted to obtain information on differential metabolites between the model group and the KSCF group. Substances with a VIP of >1 and *p*-value of <0.05 were selected as biomarkers (Figure 5). A total of 161 potential biomarkers were detected in the dorsal skin, of which 102 metabolites were up-regulated, and 59 metabolites were down-regulated (Table 1; their chemical structures are shown in Appendix A). These results suggested that KSCF might regulate these differential metabolites to relieve IMQ-induced psoriasis.

### 2.5. Metabolic Pathway Analysis

The differential metabolites were subjected to a pathway enrichment analysis to explore the potential metabolic pathways of KSCF exerting medicinal effects. Figure 6 shows the path influence diagrams of the metabolic pathway analysis. Pathways with a *p*-value of <0.05 were considered KSCF-involved pathways. The difference between the model and KSCF groups was clearly reflected in three pathways (Table 2), which are mainly related to lipid metabolism and steroidogenic activity.

Sphingolipids have structural and biological functions in the human epidermis, are importantly involved in the maintenance of the skin barrier, and regulate cellular processes, such as the proliferation, differentiation, and apoptosis of keratinocytes [23,24]. In this study, 3-O-sulfogalactosylceramide (d18:1/18:1(9Z)) and sphingosine 1-phosphate were identified as metabolites with significant differences after modeling and drug administration, suggesting that these key metabolites could be targets for KSCF against IMQ-induced psoriasis. Therefore, it was inferred that KSCF improved IMQ-induced psoriasis by regulating sphingolipid metabolism pathways.

Abnormal linoleic acid metabolism has been shown to be a key pathway in psoriasis. The combination of a moisturizer containing a linoleic acid–ceramide complex and glucocorticoids was shown to significantly improve the therapeutic efficacy of psoriasis and delay recurrence [25]. Our results confirmed that the levels of PC (14:1(9Z)/18:0) and PC (18:0/22:4(7Z,10Z,13Z,16Z)) were close to the normal group after KSCF administration. These results suggested that KSCF improved IMQ-induced psoriasis by regulating the linoleic acid metabolism pathways.

The skin has an endogenous steroidogenic capacity, and modulating local steroidogenic activity may be a new approach to treating inflammatory skin diseases [26,27]. In this study, 5α-Pregnan-20α-ol-3-one and 21-Hydroxy-5b-pregnane-3,11,20-trione were identified as metabolites with significant differences after modeling and drug administration. After KSCF administration, steroid hormone biosynthesis was similar to the normal group, suggesting that KSCF treatment affected steroid hormone biosynthesis to relieve IMQ-induced psoriasis.

Our results indicated that KSCF inhibited the inflammatory response to prevent IMQ-induced psoriasis-like skin lesions in mice by call-backing the levels of endogenous metabolites and affecting their related metabolic pathways (Figure 7). It is very likely that the flavonoid extract from *S. flavescens* also shows the therapeutic effects on psoriasis because it contains many active compounds, such as KSCF, which is worthy of further study. The deep molecular mechanism of KSCF regulating endogenous metabolites in three key pathways is also worthy to be further explored by detecting the key proteins in the pathways. It is reported that another flavonoid (kurarinone) and alkaloids (matrine and oxymarine) isolated from *S. flavescens* regulated the inflammatory response to intervene in psoriasis [7,28,29]. However, the differences in the efficacy and mechanism of these compounds in the psoriasis treatment and the relationship between the chemical structures of them and their anti-psoriasis activity are still unclear, which is worthy of further study.

## 3. Materials and Methods

### 3.1. Chemicals, Reagents, and Materials

KSCF was provided from Chengdu Pufei De Biotech Co., Ltd. (Chengdu, China). CVD was purchased from A&M Pharmaceuticals (Hong Kong, China). Hematoxylin stain was obtained from Wuhan Google Biotechnology Co., Ltd. (Wuhan, China). Vaseline for medical use was purchased from Shandong Lircon Medical Technology Co., Ltd. (Dezhou, China). IMQ was acquired from Sichuan MED-SHINE Pharmaceutical Co., Ltd. (Chengdu, China). Four percent paraformaldehyde solution was provided from Beijing lanjieke Technology Co., Ltd. (Beijing, China). IL-6, IL-8, IL-1β, IL-10, IL-22, IL-23, IL-17A, and TNF-α ELISA Kits were purchased from Jiangsu Meimian Industrial Co., Ltd. (Yancheng, China). The reference substance of KSCF (purity > 98%) was purchased from Chengdu Pufei De Biotech Co., Ltd. (Chengdu, China). High purity deionized water was purified using a Milli-Q system (Millipore, Bedford, MA, USA). HPLC-grade formic acid, methyl alcohol, and acetonitrile were acquired from Fischer Control Equipment International Co., Ltd. (Hong Kong, China). H&E Staining kits were purchased from Beijing Solarbio Science&Technology Co., Ltd. (Beijing, China). Seventy-five percent ethanol was purchased from Sinopharm Chemical Reagent Co., Ltd. (Shanghai, China). All other reagents used were of analytical reagent grade or higher.

### 3.2. Animals and Experimental Protocol

All experimental procedures in this study complied with the National Guidelines for the Care and Use of Laboratory Animals and were approved by the Animal Ethics Committee of Yunnan University of Chinese Medicine, with ethics number R062021158.

Healthy male BALB/c mice (20 ± 2 g), aged 6–8 weeks, were obtained from Sipeifu Biotechnology Co., Ltd. (Beijing, China). The mice were kept in the Laboratory of Yunnan University of Traditional Chinese Medicine (Kunming, China), following standard housing conditions. All mice were adaptively fed for one week. The mice were randomized into six groups (n = 6 mice per group), including control group (75% ethanol), model group (75% ethanol), positive group (CVD, 62.5 mg/kg), low-dose KSCF group (L-KSCF, 200 mg/kg), medium-dose KSCF group (M-KSCF, 400 mg/kg), and high-dose KSCF group (H-KSCF, 600 mg/kg). Except for the control group, mice were topically treated with 5% IMQ cream at a daily dose of 62.5 mg for a duration of 5 days on their dorsal region. KSCF and CVD were dissolved in 75% ethanol. Starting from the day of modeling, the corresponding drugs were sprayed continuously for 5 days (0.3 mL/day/mouse) in the administration group. Seventy-five percent ethanol (0.3 mL/day/mouse) was sprayed for the control group and the model group.

### 3.3. Psoriasis Area and Severity Index (PASI) Assessment

According to clinical PASI score standard, the severity of the skin inflammation was evaluated once daily, including the measurements for skin erythema, scaling, and thickening. Erythema, scaling, and thickening were scored independently on a scale from 0 to 4: 0, none; 1, slight; 2, moderate; 3, marked; 4, very marked. The cumulative score (erythema plus scaling plus thickening) served as a measure of the severity of inflammation (scale 0–12) [30].

### 3.4. Image Acquisition and Skin Sample Collection

After euthanizing the mice through carbon dioxide asphyxiation, images were captured on the dorsal side of each mouse group. Then, the dorsal skin of the mice was carefully excised using surgical scissors and divided into two sections. One section was immediately immersed in liquid nitrogen for cryopreservation, while the other section was fixed in 4% paraformaldehyde universal tissue fixative.

### 3.5. Histopathology Analyses

The dorsal skin of mice was fixed in 4% paraformaldehyde universal tissue fixative for 24 h, washed with PBS, dehydrated with gradient ethanol and transparentized with xylene, and embedded in paraffin. After hematoxylin and eosin staining, histological parameters were observed under a light microscope, and images were taken at 200× magnification.

### 3.6. Measurement of Skin Inflammatory Factors

Approximately 50 mg of injured skin from the dorsal region of each mouse were weighed and placed on ice. The surface blood stains on the skin samples were rinsed with pre-cooled normal saline solution. After air-drying the filter paper, the skin samples were promptly sectioned into pieces and transferred into a covered 2 mL centrifuge tube. The tube was then subjected to centrifugation (Centrifuge 5430R, Eppendorf, Germany) at 1500× *g* for 15 min, and subsequently, the supernatant of the homogenized tissue was collected for further utilization. The levels of cytokines IL-1β, IL-6, IL-8, IL-10, IL-17A, IL-22, IL-23, and TNF-α in the dorsal skin lesions of all groups of mice were detected according to the kit manufacturer’s instructions. The values were determined by measuring the absorbance value (OD) at 450 nm using a microplate reader (Rayto Life and Analytical Sciences Co., Ltd., Shenzhen, China).

### 3.7. Metabolomic Analysis

#### 3.7.1. Sample Pre-Treatment

Skin samples: 50 mg skin sample and a 6 mm diameter grinding bead was added to a 2 mL centrifuge tube. An amount of 400 μL of extraction solution (methanol: water = 4:1 (*v*:*v*)) was used for metabolite extraction. Samples were ground with Wonbio-96c frozen tissue grinder (Shanghai Wanbo Biotechnology Co., Ltd., Shanghai, China) for 6 min (−10 °C, 50 Hz) followed by low-temperature ultrasonic extraction for 30 min (5 °C, 40 kHz). Then, the sample was allowed to stand for 30 min at −20 °C and centrifuged for 15 min (4 °C, 3500× *g*). The supernatant was transferred to a clean tube and dried gently with nitrogen. The residues were redissolved in 200 µL of methanol for UHPLC/MS analysis.

Quality control sample: A random injection sequence was employed to detect signal fluctuation. An amount of 2 μL of skin samples were taken and thoroughly mixed. Then, the mixture was centrifuged at 3500× *g* for 15 min at 4 °C. The supernatant was collected to perform UHPLC/MS analysis.

#### 3.7.2. UHPLC/MS Analysis

UHPLC analyses were performed using Ultimate 3000 Ultra High Performance Liquid Chromatography (Thermo Fisher Scientific, San Jose, CA, USA) equipped with a Thermo Scientific Hypersil GOLD column (50 mm × 2.1 mm, 1.9 μm). The column temperature was maintained at 40 °C to obtain better sample separation. The mobile phases consisted of water containing 0.1% formic acid (A) and acetonitrile (B) at a 0.4 mL/min of flow rate. The gradient program was set as follows: 0–3.5 min, 0% B → 24.5% B; 3.5–5 min, 24.5% B → 65% B; 5–7.4 min, 65% B → 100% B; 7.4–7.6 min, 100% B → 51.5% B; 7.6–10 min, 51.5% B → 0% B.

The MS data were collected using a Thermo Scientific Q-Exactive TM hybrid quadrupole-orbitrap mass spectrometer with a heated electrospray ionization probe (Thermo Fisher Scientific, San Jose, CA, USA). MS conditions were as follows: CUR, 15 psi; Gas1 and Gas2, 50 psi; IS, 5500 V; gas temperature, 500 °C.

### 3.8. Statistical Analysis

All UHPLC/MS raw files were exported in comma-separated value (CSV) format using the Progenesis QI (Waters Corporation, Milford, CT, USA) software. All data were uploaded to XCMS online for peak alignment, normalization, and retrieval. The data were analyzed with principal component analysis (PCA) and orthogonal partial least squares discriminant analysis (OPLS-DA) using the SIMCA-P 14.1 software (Sweden, Umeå, Umetrics). The potential biomarkers were selected according to the parameters of variable importance in the projection (VIP > 1 and *p* < 0.05) from OPLS-DA. The structural information of the metabolites was identified using the HMDB databases. Finally, the metabolic pathways were searched using Kyoto Encyclopedia of Genes and Genomes (KEGG) through MetaboAnalyst 5.0 online, and the pathways with an impact value greater than 0.1 were considered KSCF-involved pathways.

All experimental data were presented as means ± S.D. Each experiment was performed in triplicate. The overall significance of the results was examined with one-way ANOVA using GraphPad Prism 5.0 (GraphPad Software, La Jolla, CA, USA). Kolmogorov–Smirnov test was used for the pre-test of the analysis of ANOVA to ensure the rationality of the statistical analysis. Student’s *t*-test delivered the *p*-values shown. The differences between the compared groups were considered statistically significant at *p* < 0.05, *p* < 0.01, *p* < 0.001, *p* < 0.0001.

## 4. Conclusions

KSCF inhibited the inflammatory response to prevent IMQ-induced psoriasis-like skin lesions in mice by call-backing the levels of 161 endogenous metabolites and affecting the three related metabolic pathways, including sphingolipid and linoleic acid metabolism and steroid hormone biosynthesis. Thus, KSCF has the potential to be developed as a topical drug for treating psoriasis.

## Figures and Tables

**Figure 1 molecules-29-02410-f001:**
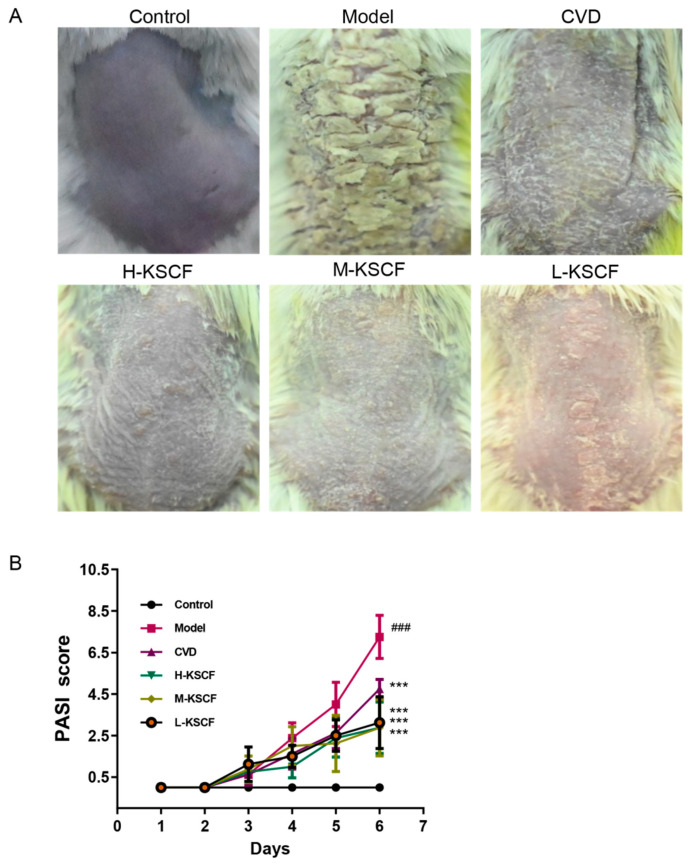
Effect of KSCF on psoriatic dermatitis induced by IMQ. KSCF mitigated skin lesions on the back (**A**) and decreased PASI values (**B**) in IMQ-induced mice. CVD, calcipotriol; H-KSCF, high-dose KSCF group; M-KSCF, medium-dose KSCF group; L-KSCF, low-dose KSCF group. ^#^ vs. control group; * vs. model group; ^###^
*p* < 0.001; ****p* < 0.001.

**Figure 2 molecules-29-02410-f002:**
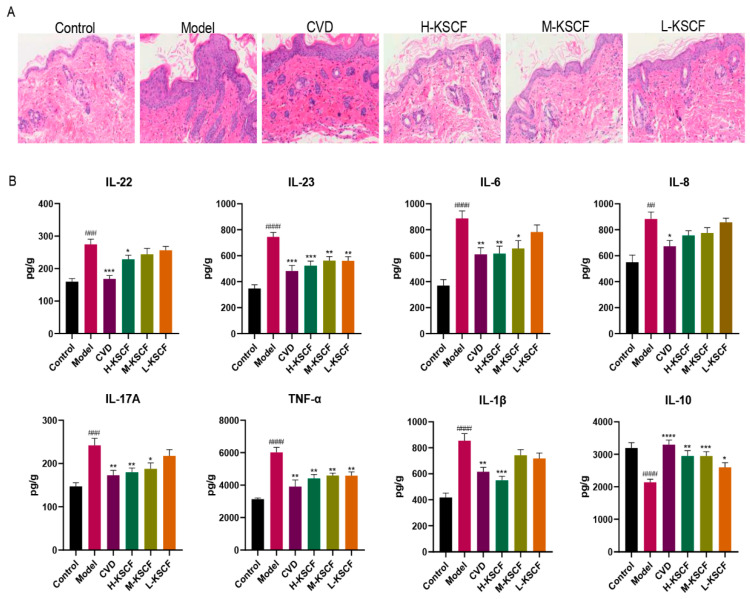
Effect of KSCF on inflammatory reactions in psoriasis−like skin lesions in mice. KSCF reduced inflammatory cell infiltration in the dorsal skin (**A**) and regulated the secretion of inflammatory cytokines (**B**). CVD, calcipotriol group; H-KSCF, high-dose KSCF group; M-KSCF, medium-dose KSCF group; L-KSCF, low-dose KSCF group. ^#^ vs. control group; * vs. model group; ^####^
*p* < 0.0001, ^###^
*p* < 0.001, ^##^
*p* < 0.01; **** *p* < 0.0001, *** *p* < 0.001, ** *p* < 0.01, * *p* < 0.05.

**Figure 3 molecules-29-02410-f003:**
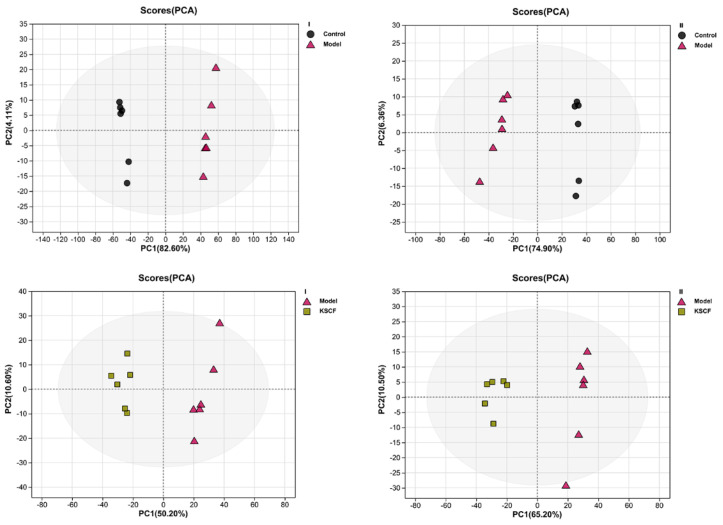
PCA score plots of psoriasis−like skin lesions in mice. I, positive ion mode; II, negative ion mode; KSCF, Kushenol F.

**Figure 4 molecules-29-02410-f004:**
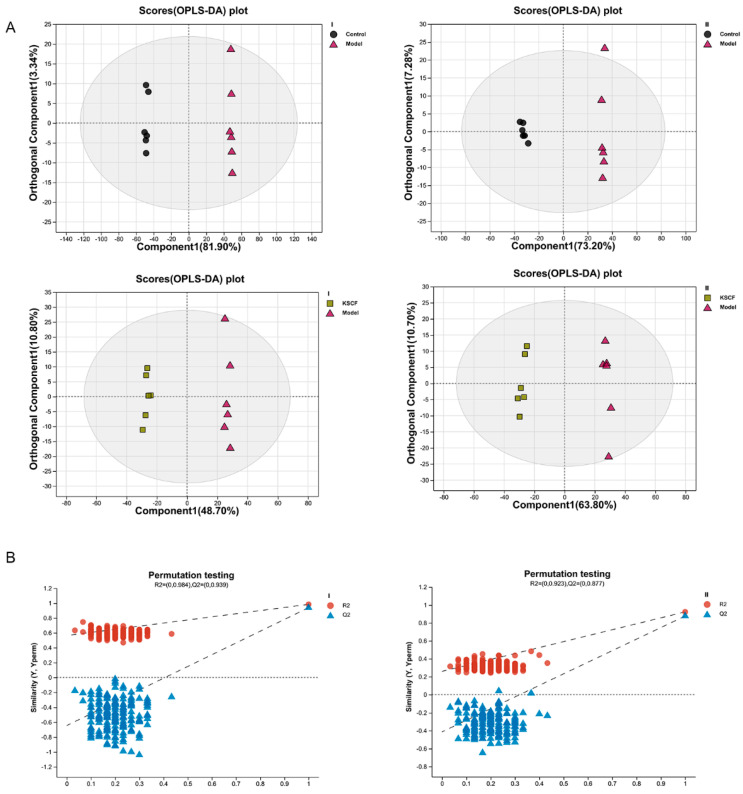
OPLS−DA score plots of psoriasis−like skin lesion samples. (**A**) OPLS−DA score plots. (**B**) Permutation test plot. I, positive ion mode; II, negative ion mode; KSCF, Kushenol F.

**Figure 5 molecules-29-02410-f005:**
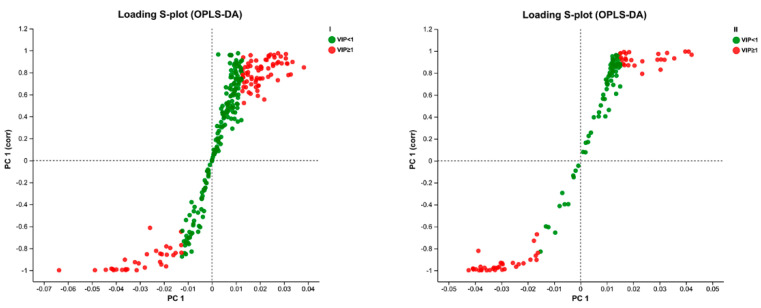
S−Plot of the model and KSCF administration group. I, positive ion mode; II: negative ion mode; red, VIP ≥ 1; green, VIP < 1.

**Figure 6 molecules-29-02410-f006:**
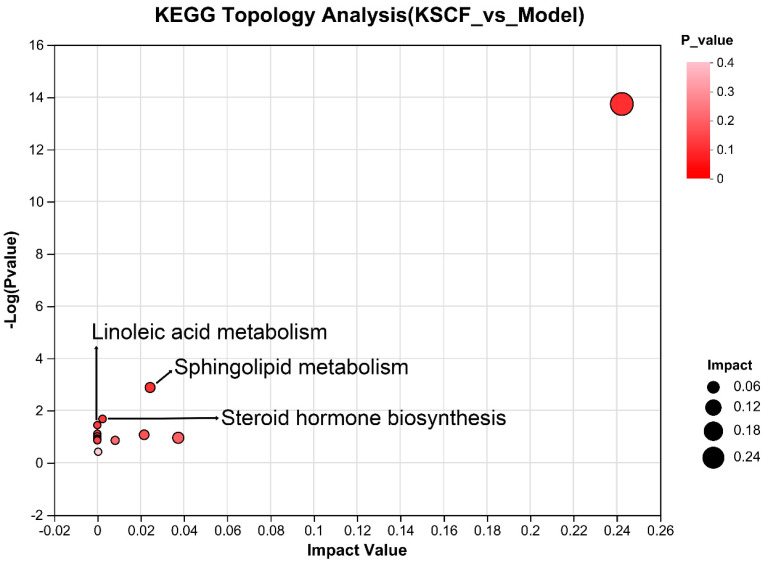
Pathway analysis of KSCF treatment.

**Figure 7 molecules-29-02410-f007:**
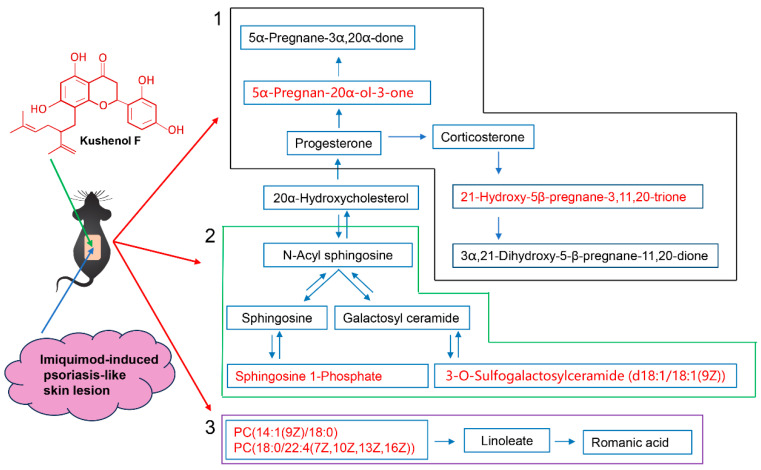
KSCF regulated endogenous metabolites in three key pathways against IMQ-induced psoriasis. 1, steroid hormone biosynthesis; 2, sphingolipid metabolism; 3, linoleic acid metabolism. Red, up.

**Table 1 molecules-29-02410-t001:** Differential metabolites identified from psoriatic skin lesion samples.

Retention Time (min)	Molecular Weight (Da)	VIP	Potential Biomarker	Formula	Change Trend
ESI−					
4.843	308.124	2.604	Glutaminyltyrosine	C_14_H_19_N_3_O_5_	DOWN
6.468	453.202	2.054	2-Hydroxy-imipramine glucuronide	C_25_H_32_N_2_O_7_	UP
6.183	581.300	2.676	Serylarginine	C_9_H_19_N_5_O_4_	UP
2.626	263.076	1.478	3,4,5-trihydroxy-6-[(2-methylpropanoyl)oxy]oxane-2-carboxylic acid	C_10_H_16_O_8_	DOWN
2.821	374.095	2.080	7-Hydroxygliclazide	C_15_H_21_N_3_O_4_S	DOWN
2.829	299.070	1.075	Methionyl-Glutamate	C_10_H_18_N_2_O_5_S	DOWN
4.270	315.163	2.005	3-[8-hydroxy-2-methyl-2-(4-methylpent-3-en-1-yl)-2*H*-chromen-5-yl]propanoic acid	C_19_H_24_O_4_	DOWN
4.791	165.059	1.301	3-Mercapto-2-methyl-1-butanol	C_5_H_12_OS	DOWN
6.240	480.249	2.770	Dynorphin A (6–8)	C_18_H_37_N_9_O_4_	UP
6.278	457.197	2.666	Deoxypyridinoline	C18H28N_4_O_7_	UP
6.308	469.161	2.372	Melleolide B	C_24_H_32_O_7_	UP
6.308	667.171	1.090	Hydroxychlorpromazine	C_17_H_19_ClN_2_OS	UP
6.347	455.181	2.248	Armillane	C_23_H_32_O_7_	UP
6.378	437.170	2.123	Nevskin	C_24_H_32_O_5_	UP
6.431	177.023	1.656	3,3′-Thiobispropanoic acid	C_6_H_10_O_4_S	UP
6.442	295.031	2.518	Cis,trans-5′-Hydroxythalidomide	C_13_H_10_N_2_O_5_	UP
6.456	421.175	2.033	Euglobal IIb	C_23_H_30_O_5_	UP
6.462	465.202	1.909	Homofukinolide	C_25_H_34_O_6_	UP
6.493	486.187	1.533	4-Hydroxyvalsartan	C_24_H_29_N_5_O_4_	UP
5.583	451.185	1.071	7-[(2,6-dihydroxy-2,5,5,8a-tetramethyl-decahydronaphthalen-1-yl)methoxy]-6-hydroxy-2H-chromen-2-one	C_24_H_32_O_6_	UP
6.601	467.217	2.471	Colupdox a	C_25_H_36_O_6_	UP
6.630	553.297	1.078	Phytolaccinic acid	C_31_H_48_O_6_	DOWN
6.969	492.117	2.575	2-amino-4-({1-[(carboxymethyl)-C-hydroxycarbonimidoyl]-2-[(2,3-dihydroxy-1-phenylpropyl)sulfanyl]ethyl}-C-hydroxycarbonimidoyl)butanoic acid	C_19_H_27_N_3_O_8_S	UP
7.023	609.361	1.208	Lactapiperanol C	C_16_H_26_O_4_	DOWN
7.434	810.583	1.023	PE(18:0/20:0)	C_43_H_86_NO_8_P	DOWN
7.480	606.500	2.048	1,1′-(1,4-Dihydro-4-nonyl-3,5-pyridinediyl)bis[1-dodecanone]	C_38_H_69_NO_2_	DOWN
8.936	453.202	1.579	Colupox a	C_25_H_36_O_5_	UP
8.915	701.421	1.880	PG(a-13:0/a-15:0)	C_34_H_67_O_10_P	UP
8.846	465.314	1.130	(2β,3α,9α,24R)-Ergosta-7,22-diene-2,3,9-triol	C_28_H_46_O_3_	DOWN
7.712	865.522	1.182	PG(18:1(11Z)/22:6(4Z,7Z,10Z,13Z,16Z,19Z))	C_46_H_77_O_10_P	DOWN
7.365	820.589	1.091	PE(18:1(11Z)/22:1(13Z))	C_45_H_86_NO_8_P	DOWN
7.157	463.293	2.768	Palmitoyl glucuronide	C_22_H_42_O_7_	DOWN
7.057	361.174	0.993	Perindoprilat	C_17_H_28_N_2_O_5_	DOWN
6.541	379.218	2.167	3′-Hydroxystanozolol	C_21_H_32_N_2_O_2_	DOWN
6.475	439.185	2.144	Lucidone A	C_24_H_34_O_5_	UP
6.456	423.154	2.463	5,7-dihydroxy-4-(1-hydroxypropyl)-6-(3-methylbut-2-en-1-yl)-8-(2-methylbutanoyl)-2H-chromen-2-one	C_22_H_28_O_6_	UP
6.451	455.217	2.656	Cavipetin D	C_25_H_38_O_5_	UP
6.431	475.163	2.153	α-CEHC glucuronide	C_22_H_30_O_10_	UP
6.378	500.167	2.330	2-amino-4-({1-[(carboxymethyl)-*C*-hydroxycarbonimidoyl]-2-{[4-(2,6-dihydroxyphenyl)-3-hydroxy-2-methylbutan-2-yl]sulfanyl}ethyl}-*C*-hydroxycarbonimidoyl)butanoic acid	C_21_H_31_N_3_O_9_S	UP
6.371	453.165	2.610	Diosbulbin H	C_23_H_30_O_7_	UP
6.354	439.186	1.945	1α-O-Methylquassin	C_23_H_32_O_6_	UP
6.347	507.174	2.465	L-Nicotianine	C_10_H_12_N_2_O_4_	UP
6.284	520.193	2.082	Hydroxynefazodone	C_25_H_32_C_l_N_5_O_3_	UP
6.240	452.218	1.926	Chondroitin sulfate E (GalNAc4,6diS-GlcA), precursor 5a	C_7_H_9_N_3_O	UP
6.222	369.211	1.248	3-keto-Digoxigenin	C_23_H_32_O_5_	DOWN
6.218	476.279	1.123	LysoPE(0:0/18:2(9Z,12Z))	C_23_H_44_NO_7_P	DOWN
6.134	457.160	1.105	Valechlorin	C_22_H_31_C_l_O_8_	UP
5.718	371.226	2.685	Digoxigenin	C_23_H_34_O_5_	DOWN
5.114	348.156	1.934	(+)-O-Methylarmepavine	C_20_H_25_NO_3_	DOWN
5.065	329.179	2.313	12,20-Dioxo-leukotriene B4	C_20_H_28_O_5_	DOWN
4.974	357.211	1.896	17-Hydroxy-E4-neuroprostane	C_22_H_32_O_5_	DOWN
4.589	357.222	1.037	Kinetensin 1-3	C_15_H_30_N_6_O_4_	UP
2.829	365.143	1.110	Valproic acid glucuronide	C_14_H_24_O_8_	DOWN
2.814	296.106	1.004	7-Aminonitrazepam	C_15_H_13_N_3_O	DOWN
1.685	295.040	1.398	*N*-Acetyldjenkolic acid	C_9_H_16_N_2_O_5_S_2_	UP
1.527	183.044	1.218	Mevalonic acid	C_6_H_12_O_4_	UP
6.371	383.161	1.501	21-Hydroxy-5b-pregnane-3,11,20-trione	C_21_H_30_O_4_	DOWN
6.278	581.263	1.973	*S*-(3-Methylbutanoyl)-dihydrolipoamide-*E*	C_13_H_25_NO_2_S_2_	UP
8.922	689.351	1.131	Zeranol	C_18_H_26_O_5_	UP
ESI+					
6.444	453.200	3.257	Pro Ser Ser Tyr	C_20_H_28_N_4_O_8_	UP
4.727	828.534	0.976	3-*O*-Sulfogalactosylceramide (d18:1/18:1(9Z))	C_42_H_79_NO_11_S	DOWN
3.285	404.222	1.210	Ser Ala Ala Arg	C_15_H_29_N_7_O_6_	UP
4.253	317.177	1.919	Capillartemisin A	C_19_H_24_O_4_	DOWN
5.069	331.193	2.827	16-Hydroxy-4-oxoretinoic acid	C_20_H_26_O_4_	DOWN
6.224	371.225	1.960	17-phenyl trinor-13,14-dihydro Prostaglandin A2	C_23_H_30_O_4_	DOWN
6.254	395.252	1.328	(3*R*, 6′*Z*)-3,4-Dihydro-8-hydroxy-3-(6-pentadecenyl)-1H-2-benzopyran-1-one	C_24_H_36_O_3_	UP
6.304	565.350	1.002	PG(22:2(13Z,16Z)/0:0)	C_28_H_53_O_9_P	UP
6.324	313.077	2.457	Inosine-5′-carboxylate	C_11_H_12_N_4_O_7_	UP
7.031	626.533	1.139	Muricatenol	C_37_H_68_O_6_	UP
6.338	301.077	2.967	Temazepam	C_16_H_13_C_l_N_2_O_2_	UP
1.543	286.157	2.191	Tryptophyl-Valine	C_16_H_21_N_3_O_3_	UP
3.194	298.150	1.266	Histidinyl-Threonine	C_10_H_16_N_4_O_4_	DOWN
3.319	281.119	1.191	6-hydroxy-7E,9E-Octadecadiene-11,13,15,17-tetraynoic acid	C_18_H_16_O_3_	UP
4.064	534.323	1.347	Cys Lys Lys Arg	C_21_H_43_N_9_O_5_S	DOWN
4.235	255.129	1.186	Isoleucyl-Threonine	C_10_H_20_N_2_O_4_	DOWN
4.305	622.377	1.446	Tyr Arg Lys Arg	C_27_H_47_N_11_O_6_	DOWN
4.474	277.124	1.256	Isobutyrylcarnitine	C_11_H_22_NO_4_^+^	UP
4.604	386.246	1.473	Pseudoginsenoside RT3	C_41_H_70_O_13_	DOWN
4.654	549.172	1.514	Myricatomentoside I	C_26_H_32_O_10_	DOWN
4.987	359.225	1.475	7,8-epoxy-17S-HDHA	C_22_H_30_O_4_	DOWN
5.118	696.452	1.502	PA(16:1(9Z)/15:0)	C_34_H_65_O_8_P	DOWN
5.170	828.533	1.338	PE(MonoMe(11,3)/MonoMe(11,3))	C_43_H_74_NO_10_P	DOWN
5.994	242.134	0.972	Lysyl-Methionine	C_11_H_23_N_3_O_3_S	UP
6.016	548.287	1.511	11-Hydroxyprogesterone 11-glucuronide	C_27_H_38_O_9_	UP
6.082	465.259	1.132	(−)-Jolkinol B	C_29_H_36_O_5_	UP
6.136	502.163	2.001	3-(3-hydroxyphenyl)-2-(4-hydroxyphenyl)-5-[(*E*)-2-(3hydroxyphenyl)ethenyl]-2,3-dihydro-1 benzofuran-6-ol	C_28_H_22_O_5_	UP
6.149	467.275	1.316	1,25-Dihydroxyvitamin D3-26,23-lactone	C_27_H_40_O_5_	UP
6.205	363.225	1.097	5α-Pregnan-20α-ol-3-one	C_21_H_34_O_2_	UP
6.224	453.221	1.813	Australigenin	C_27_H_42_O_4_	UP
6.283	534.221	1.608	Cyclocalopin D	C_23_H_32_O_13_	UP
6.297	365.241	1.267	Adipostatin A	C_21_H_36_O_2_	UP
6.324	574.253	1.565	Glaucarubolone 15-*O*-β-d-glucopyranoside	C_26_H_36_O_13_	UP
6.360	427.276	1.508	Asymmetric dimethylarginine	C_8_H_18_N_4_O_2_	UP
6.360	455.179	3.227	Gly Glu Ser Tyr	C_19_H_26_N_4_O_9_	UP
6.367	441.200	4.192	Gly Thr Tyr Thr	C_19_H_28_N_4_O_8_	UP
6.416	435.294	2.760	Alanyl-Lysine	C_9_H_19_N_3_O_3_	UP
6.437	527.345	2.800	Pro Val Arg Arg	C_22_H_42_N_10_O_5_	UP
6.451	355.125	2.024	Meta-*O*-Dealkylated flecainide	C_15_H_19_F_3_N_2_O_3_	UP
6.472	463.182	2.851	PA(8:0/8:0)	C_19_H_37_O_8_P	UP
6.479	441.200	3.151	(9*S*,10*S*)-10-hydroxy-9-(phosphonooxy)octadecanoate	C_18_H_37_O_7_P	UP
6.500	714.509	1.058	PA(18:1(9Z)/18:3(9Z,12Z,15Z))	C_39_H_69_O_8_P	UP
6.521	523.411	2.648	Panaxydol linoleate	C_35_H_54_O_3_	DOWN
6.604	411.225	1.903	Pro Pro Thr Pro	C_19_H_30_N_4_O_6_	UP
6.646	722.519	1.386	PE(18:4(6Z,9Z,12Z,15Z)/P-18:1(11Z))	C_41_H_72_NO_7_P	UP
6.660	678.492	1.312	PE(18:4(6Z,9Z,12Z,15Z)/P-16:0)	C_39_H_70_NO_7_P	UP
6.674	634.465	1.471	Tsugarioside B	C_37_H_60_O_7_	UP
6.856	670.523	1.605	PE(15:0/18:0)	C_38_H_76_NO_8_P	UP
7.010	904.712	1.184	PC(o-20:0/22:0)	C_50_H_102_NO_7_P	UP
7.066	407.153	1.119	3b,16a-Dihydroxyandrostenone sulfate	C_19_H_28_O_6_S	UP
8.944	425.205	4.129	*O*-Desmethylcarvedilol	C_23_H_24_N_2_O_4_	UP
9.287	423.189	2.131	Cys Val Thr Thr	C_16_H_30_N_4_O_7_S	UP
8.685	895.754	1.939	TG(18:1(11Z)/22:4(7Z,10Z,13Z,16Z)/18:3(9Z,12Z,15Z))	C_61_H_102_O_6_	DOWN
6.940	539.360	1.721	3-methyl-4-(methylamino)-1,2-diphenylbutan-2-ol	C_18_H_23_NO	UP
6.870	626.496	1.675	PE(15:0/P-16:0)	C_36_H_72_NO_7_P	UP
6.870	614.353	1.399	LysoPC(22:2(13Z,16Z))	C_30_H_58_NO_7_P	UP
6.849	714.550	1.546	PC(14:1(9Z)/18:0)	C_40_H_78_NO_8_P	UP
6.828	802.604	1.578	PC(18:0/22:4(7Z,10Z,13Z,16Z))	C_48_H_88_NO_8_P	UP
6.737	696.496	1.312	CL(i-12:0/18:2(9Z,11Z)/i-18:0/i-19:0)	C_76_H_144_O_17_P_2_	UP
6.730	740.523	1.185	CL(18:0/18:0/16:1(9Z)/22:6(4Z,7Z,10Z,13Z,16Z,19Z))	C_83_H_148_O_17_P_2_	UP
6.646	840.578	1.939	PG(18:0/22:4(7Z,10Z,13Z,16Z))	C_46_H_83_O_10_P	DOWN
6.604	163.076	1.060	4-Hydroxy-2,6,6-trimethyl-3-oxo-1,4-cyclohexadiene-1-carboxaldehyde	C_10_H_12_O_3_	UP
6.590	295.066	1.816	2-hydroxy-3-[3-(3-methylbut-2-en-1-yl)-4-(sulfooxy)phenyl]propanoic acid	C_14_H_18_O_7_S	UP
6.583	459.187	2.136	Ser Glu Ser His	C_17_H_26_N_6_O_9_	UP
6.542	355.194	1.101	Nigakihemiacetal B	C_22_H_30_O_6_	DOWN
6.493	764.516	1.715	CL(16:0/18:0/18:0/22:5(7Z,10Z,13Z,16Z,19Z))	C_83_H_152_O_17_P_2_	DOWN
6.472	700.493	1.413	PA(15:0/18:2(9Z,12Z))	C_36_H_67_O_8_P	UP
6.472	385.136	3.206	2′,7-Dihydroxy-4′-methoxy-8-prenylflavan	C_21_H_24_O_4_	UP
6.465	351.130	1.754	N-Ribosylhistidine	C_11_H_17_N_3_O_6_	UP
6.451	521.396	1.727	Ginsenoyne A linoleate	C_35_H_52_O_3_	DOWN
6.388	285.081	1.654	(2*S*)-2-hydrazinyl-3-(4-hydroxy-3-methoxyphenyl)-2-methylpropanoic acid	C_11_H_16_N_2_O_4_	DOWN
6.381	439.184	3.069	Gln Glu Tyr	C_19_H_26_N_4_O_8_	UP
6.345	375.115	1.271	Cyclic *N*-Acetylserotonin glucuronide	C_18_H_20_N_2_O_8_	DOWN
6.338	361.135	2.023	7-Hydroxy-6-methyl-8-ribityl lumazine	C_12_H_16_N_4_O_7_	UP
6.324	538.255	1.523	Rubraflavone B	C_30_H_34_O_5_	UP
6.311	618.283	2.136	3′-Deaminofusarochromanone	C_15_H_19_NO_4_	UP
6.297	424.221	2.812	Sphingosine 1-phosphate	C_18_H_38_NO_5_P	UP
6.290	508.280	2.746	LysoPE(0:0/22:6(4Z,7Z,10Z,13Z,16Z,19Z))	C_27_H_44_NO_7_P	UP
6.283	510.296	4.312	LysoPE(0:0/22:5(7Z,10Z,13Z,16Z,19Z))	C_27_H_46_NO_7_P	UP
6.276	583.277	3.375	(4*E*)-6-hydroxy-1,7-diphenylhept-4-en-3-one	C_19_H_20_O_2_	UP
6.241	451.279	0.944	PA(19:1(9Z)/0:0)	C_22_H_43_O_7_P	UP
6.231	498.259	3.159	PE(18:3/0:0)	C_23_H_42_NO_7_P	UP
6.199	437.264	1.065	1-Oleoyl Lysophosphatidic Acid (sodium salt)	C_21_H_41_O_7_P	UP
6.186	175.123	2.811	α-Methyltryptamine (AMT)	C_11_H_14_N_2_	UP
6.174	569.298	3.370	11-α-*O*-β-d-Glucopyranosyl-16α-*O*-methylneoquassin	C_29_H_44_O_11_	UP
6.075	487.359	1.085	Lys Ile Val Lys	C_23_H_46_N_6_O_5_	UP
6.005	618.340	1.903	Pentasine	C_30_H_49_N_6_O_10_^+^	UP
6.000	349.209	1.127	Oryzalexin E	C_20_H_32_O_2_	UP
5.554	183.082	1.526	2-Methyl-4-propyl-1,3-oxathiane	C_8_H_16_OS	UP
5.112	740.479	1.500	PS(14:0/20:3(8Z,11Z,14Z))	C_40_H_72_NO_10_P	DOWN
5.006	240.142	0.993	Isopropyl β-d-glucoside	C_9_H_18_O_6_	UP
4.945	336.130	0.976	Gynocardin	C_12_H_17_NO_8_	DOWN
4.897	352.161	1.890	(+/−)-Rollipyrrole	C_16_H_20_N_2_O_3_	DOWN
4.615	338.145	1.145	*N*-Acetyl-3-hydroxyprocainamide	C_15_H_23_N_3_O_3_	DOWN
4.391	233.097	1.094	L-Targinine	C_7_H_16_N_4_O_2_	UP
3.611	402.242	1.022	*N*-Didesmethylmifepristone (RU 42848)	C_27_H_31_NO_2_	DOWN
3.396	261.150	1.327	Methyl dihydrophaseate	C_16_H_24_O_5_	UP
3.222	229.123	1.660	Blennin B	C_15_H_20_O_4_	UP
2.566	267.140	1.527	(*S*)-17-Hydroxy-9,11,13,15-octadecatetraynoic acid	C_18_H_20_O_3_	UP
1.360	217.134	1.518	Girgensonine	C_13_H_16_N_2_O	DOWN
0.013	300.208	1.040	2-hydroxydesipramine	C_18_H_22_N_2_O	UP
6.800	524.283	1.270	11-β-Hydroxyandrosterone-3-glucuronide	C_25_H_38_O_9_	UP

**Table 2 molecules-29-02410-t002:** Metabolic pathway of psoriatic skin lesion samples.

No.	Pathway	*p*-Value	Details
1	Sphingolipid metabolism	0.001	KEGG
2	Linoleic acid metabolism	0.036	KEGG
3	Steroid hormone biosynthesis	0.021	KEGG

## Data Availability

Data are available upon request.

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
