# Peer review of "Metabolic Profiling for Unveiling Mechanisms of Kushenol F against Imiquimod-Induced Psoriasis with UHPLC/MS Analysis"

_molecules, 2024, doi:10.3390/molecules29112410_

Round 1
Reviewer 1 Report
Comments and Suggestions for Authors
In this manuscript, Chen et al. have presented an interesting study on Metabolic Profiling for Unveiling Mechanisms of Kushenol F against Imiquimod-induced Psoriasis. The study gives insight into the influence of Kushenol F on the metabolism in a psoriasis mouse model and also presents the influence on the expression of psoriasis key-cytokines. The manuscript is well written and has a clear structure. However, I have a few comments on the manuscript that should be considered in the revision process.
Comments:
Line 43: Please add a reference which represents the epidemiological situation of psoriasis described in this sentence.
Line 51: Please note that the expulsion of evil wind originates from the teachings of traditional Chinese medicine.
Line 149: Please establish the abbreviation for UPLHC/MS once again. Abbreviations should be defined the first time they appear in each of three sections: the abstract; the main text; the first figure or table. Please check this throughout the whole manuscript.
Line 179: Please add Ethanol 75% and H&E as well as information on the corresponding sources.
Lines 233 + 247: Please add information on the centrifuge used. G-values instead of rpm are suggested.
Line 280: Was an ANOVA justified by pre-tests testing the normal distribution? Which ANOVA post-hoc tests delivered the p values shown?
Comments on the Quality of English LanguageOnly minor language editing required.
Author Response
Comments:
Line 43: Please add a reference which represents the epidemiological situation of psoriasis described in this sentence.
Response: Thanks for your kind and professional suggestion. The reference which represents the epidemiological situation of psoriasis described in this sentence has been supplemented (Int J Mol Sci. 2023;24(17):13313) (Line 43).
Line 51: Please note that the expulsion of evil wind originates from the teachings of traditional Chinese medicine.
Response: Thank you for this suggestion, we have noted that the expulsion of evil wind originates from the teachings of traditional Chinese medicine, which has been supplemented in our manuscript (Lines 56-58).
Line 149: Please establish the abbreviation for UPLHC/MS once again. Abbreviations should be defined the first time they appear in each of three sections: the abstract; the main text; the first figure or table. Please check this throughout the whole manuscript.
Response: As described by the reviewer, we have checked this throughout the whole manuscript, and establish the abbreviation of UHPLC/MS again in the main text (Lines 114-115).
Line 179: Please add Ethanol 75% and H&E as well as information on the corresponding sources.
Response: Thank you for your kind suggestions. Ethanol 75% and H&E as well as information on the corresponding sources have been supplemented in our manuscript (Lines 221-224).
Lines 233 + 247: Please add information on the centrifuge used. G-values instead of rpm are suggested.
Response: As described by the reviewer, the information of the centrifuge used has been supplemented (Lines 267-268). The rpm in our manuscript has been replaced by G-values (Lines 268, 282, 286).
Line 280: Was an ANOVA justified by pre-tests testing the normal distribution? Which ANOVA post-hoc tests delivered the p values shown?
Response: Thanks for your kind and professional suggestion. In the manuscript, Kolmogorov Smirnov test is used for the pre-test of the analysis of ANOVA to ensure the rationality of the statistical analysis. Student's t-test delivered the p values shown. These have been supplemented in our manuscript (Lines 316-317).

Reviewer 2 Report
Comments and Suggestions for Authors
Kushenol F (KSCF) is a natural isopentenyl flavonoid extracted from the root of Sophora flavescens Alt. The effect and mechanism of KSCF on imiquimod (IMQ)-induced psoriasis-like skin lesions in mice was investigated. A mouse model of psoriasis was induced with 5% IMQ for 5 days, and the mice were given KSCF dermally for 5 days. Changes in skin morphology, the psoriasis area, severity index (PASI), and inflammatory factors of psoriasis-like skin lesions were evaluated. Metabolites in the psoriasis-like skin lesions were analyzed by ultra-high-performance liquid chromatography/mass spectrometry (UHPLC/MS), followed by multivariate statistical analysis to identify the differential metabolites and metabolic pathway. The results of the present study confirmed that KSCF significantly reduced PASI scores, epidermal thickening, and epidermal cell proliferation and differentiation. KSCF also reduced the levels of interleukin (IL)-1β, IL-6, IL-8, IL-17A, IL-22, IL-23, and tumor necrosis factor (TNF)-α in the injured skin tissues, while increased IL-10 content. KSCF significantly regulated metabolites in the skin samples, and a total of 161 significant metabolites were identified. These differential metabolites involved sphingolipid and linoleic acid metabolism and steroid hormone biosynthesis. Collectively, KSCF inhibited the inflammatory response to prevent IMQ-induced psoriasis-like skin lesions in mice by call-backing the levels of 161 endogenous metabolites and affected their related metabolic pathways. KSCF has the potential to be developed as a topical drug for treating psoriasis symptoms.
It is well design and the result is interesting. The manuscript has been revised and the quality has been improved. However, there still have minor revision need to improve.
1. The expression should be checked in the manuscript.
2. In introduction part. The inflammation marker should be introduced.
3. The inflammatary cytokine and cytometry shold be measured. Please refer this reference(Cancer Research, 2020, 80(12): 2564-2574.Cell reports 41(11):111804).
4. The KEGG pathway about Kushenol F metabolites is insufficient.
5. There should be significant different analysis in the table and the main text.
6. The reference should be updated.
Comments on the Quality of English LanguageKushenol F (KSCF) is a natural isopentenyl flavonoid extracted from the root of Sophora flavescens Alt. The effect and mechanism of KSCF on imiquimod (IMQ)-induced psoriasis-like skin lesions in mice was investigated. A mouse model of psoriasis was induced with 5% IMQ for 5 days, and the mice were given KSCF dermally for 5 days. Changes in skin morphology, the psoriasis area, severity index (PASI), and inflammatory factors of psoriasis-like skin lesions were evaluated. Metabolites in the psoriasis-like skin lesions were analyzed by ultra-high-performance liquid chromatography/mass spectrometry (UHPLC/MS), followed by multivariate statistical analysis to identify the differential metabolites and metabolic pathway. The results of the present study confirmed that KSCF significantly reduced PASI scores, epidermal thickening, and epidermal cell proliferation and differentiation. KSCF also reduced the levels of interleukin (IL)-1β, IL-6, IL-8, IL-17A, IL-22, IL-23, and tumor necrosis factor (TNF)-α in the injured skin tissues, while increased IL-10 content. KSCF significantly regulated metabolites in the skin samples, and a total of 161 significant metabolites were identified. These differential metabolites involved sphingolipid and linoleic acid metabolism and steroid hormone biosynthesis. Collectively, KSCF inhibited the inflammatory response to prevent IMQ-induced psoriasis-like skin lesions in mice by call-backing the levels of 161 endogenous metabolites and affected their related metabolic pathways. KSCF has the potential to be developed as a topical drug for treating psoriasis symptoms.
It is well design and the result is interesting. The manuscript has been revised and the quality has been improved. However, there still have minor revision need to improve.
1. The expression should be checked in the manuscript.
2. In introduction part. The inflammation marker should be introduced.
3. The inflammatary cytokine and cytometry shold be measured. Please refer this reference(Cancer Research, 2020, 80(12): 2564-2574.Cell reports 41(11):111804).
4. The KEGG pathway about Kushenol F metabolites is insufficient.
5. There should be significant different analysis in the table and the main text.
6. The reference should be updated.
Author Response
Comments:
- The expression should be checked in the manuscript.
Response: Thank you for your kind suggestions. The expression has been checked in the manuscript.
- In introduction part. The inflammation marker should be introduced.
Response: As described by the reviewer, the inflammation marker has been introduced in the introduction part (Lines 43-47).
- The inflammatary cytokine and cytometry shold be measured. Please refer this reference(Cancer Research, 2020, 80(12): 2564-2574.Cell reports 41(11):111804).
Response: Thank you very much for the very good and novel idea from reviewer. Although the detection of inflammatary cytometry was not performed due to the lack of sufficient biological samples and high-dimensional time-of-flight mass cytometry (which will be conducted in the future studies), the levels of inflammatary cytokines IL-1β, IL-6, IL-8, IL-10, IL-17A, IL-22, IL-23, and TNF-α in the dorsal skin lesions of all groups of mice were detected by ELISA kits, which indicates that KSCF might stimulate the secretion of anti-inflammatory cytokines while inhibiting the release of pro-inflammatory cytokines, thereby alleviating IMQ-induced psoriasis. In addition, these innovative research from the references provided by reviewer have been cited in our manuscript (Lines 47-48).
- The KEGG pathway about Kushenol F metabolites is insufficient.
Response: Thank you very much for the very good and novel idea from reviewer. Our results indicate that the molecular mechanism of KSCF regulating endogenous metabolites against IMQ-induced psoriasis-like skin lesions involved three key pathways. The deep molecular mechanism of KSCF regulating endogenous metabolites in three key pathways is worthy to be further explored by detecting the key proteins in the pathways. Unfortunately, the biological samples including skin and serum in this study have been stored for too long (about 18 months), and the constituents in these samples may be significantly changed. Thus, these samples may not be used for further experiments, which will be meet in future research. This shortcoming has been supplemented in our manuscript (Lines 190-192).
- There should be significant different analysis in the table and the main text.
Response: We agree with what the reviewer described. The description of the table in the main text has been revised to show a significant difference (Lines 159-161).
- The reference should be updated.
Response: The references have been updated (Lines 358-359, 371-372, 394-395).

Reviewer 3 Report
Comments and Suggestions for Authors
In this study, the effect and mechanism of Kushenol F (KSCF) on imiquimod-induced psoriasis-like skin lesions in mice were investigated. The changes in skin morphology, the psoriasis area, severity index, and inflammatory factors of psoriasis-like skin lesions were evaluated. Metabolites in the psoriasis-like skin lesions were analyzed by ultra-high-performance liquid chromatography/mass spectrometry, followed by multivariate statistical analysis to identify the differential metabolites and metabolic pathway. The results of present work are beneficial for the further study on the treatment of psoriasis by KSCF. The manuscript may be accepted for publication after major revisions.
1. Please explain why the extracts, such as the flavonoid extract, from Sophora flavescens haven’t been tested.
2. In the introduction section, the studies on the metabolic analysis of psoriasis should be introduced. The key innovations (if any) on the methodology of present work can be emphasized.
3. The graphics in Figure 1 can be rearranged. The graphics of Figure 1A can be displayed in two rows. The resolution of figures can be further improved.
4. The total ion chromatograms of the LC-MS can be provided, at least as supplementary materials. If possible, the chemical structures of the key metabolites can also be provided as supplementary materials.
5. The Figure 7 should be moved to the Results and discussion section. The shortcomings of present work can be emphasized in the conclusion section.
6. The “KSCF regulated endogenous metabolites in 3 key pathways”, why further experiments haven’t been performed to verified these results?
7. The “Institutional Review Board Statement” should be revised, as this study has involved animal experiments.
8. Table 2 can be deleted, and describe the content in the main text.
9. The abbreviation form “S. flavescens” can be used after its full name “Sophora flavescens Alt” has been given.
10. There should be a space between the numeral and unit.
Author Response
Comments:
Reviewer 3:
In this study, the effect and mechanism of Kushenol F (KSCF) on imiquimod-induced psoriasis-like skin lesions in mice were investigated. The changes in skin morphology, the psoriasis area, severity index, and inflammatory factors of psoriasis-like skin lesions were evaluated. Metabolites in the psoriasis-like skin lesions were analyzed by ultra-high-performance liquid chromatography/mass spectrometry, followed by multivariate statistical analysis to identify the differential metabolites and metabolic pathway. The results of present work are beneficial for the further study on the treatment of psoriasis by KSCF. The manuscript may be accepted for publication after major revisions.
- Please explain why the extracts, such as the flavonoid extract, from Sophora flavescens haven’t been tested.
Response: Thank you very much for the very good and novel idea from reviewer. Our results indicate that KSCF (a single compound from Sophora flavescens) inhibited the inflammatory response to prevent IMQ-induced psoriasis-like skin lesions in mice by call-backing the levels of 161 endogenous metabolites and affected their related metabolic pathways. As described by the reviewer, it is very likely that the flavonoid extract from Sophora flavescens also shows the therapeutic effects on psoriasis because it contains many active compounds such as KSCF, which is worthy of further study. These have been supplemented in our manuscript (Lines 185-190).
- In the introduction section, the studies on the metabolic analysis of psoriasis should be introduced. The key innovations (if any) on the methodology of present work can be emphasized.
Response: As described by the reviewer, the studies on the metabolic analysis of psoriasis has been introduced in the introduction section (Lines 65-69). In the present study, the mature and stable metabolic analytical methods were used for uncovering the novel effects and mechanisms of KSCF against psoriasis, and the key innovations on the methodology are not emphasized.
- The graphics in Figure 1 can be rearranged. The graphics of Figure 1A can be displayed in two rows. The resolution of figures can be further improved.
Response: Thank you for your kind suggestions. The graphics of Figure 1A has been rearranged and the resolution of the number has been further improved.
- The total ion chromatograms of the LC-MS can be provided, at least as supplementary materials. If possible, the chemical structures of the key metabolites can also be provided as supplementary materials.
Response: Thank you very much for the very good and novel idea from reviewer. The total ion chromatograms of the LC-MS and chemical structures of the key metabolites have been provided as supplementary materials (Supplementary Figures S1 and S2). These have been supplemented in our manuscript (Lines 114-120, 147).
- The Figure 7 should be moved to the Results and discussion section. The shortcomings of present work can be emphasized in the conclusion section.
Response: We agree with what the reviewer described. The Figure 7 has been moved to the Results and discussion section (Line 187). Furthermore, the shortcomings of present work can be emphasized in the Results and discussion section (Lines 187-197): It is very likely that the flavonoid extract from S. flavescens also shows the therapeutic effects on psoriasis because it contains many active compounds such as KSCF, which is worthy of further study. Besides, the deep molecular mechanism of KSCF regulating endogenous metabolites in three key pathways is also worthy to be further explored by detecting the key proteins in the pathways. In addition, it is reported that another flavonoid (kurarinone) and alkaloids (matrine and oxymarine) isolated from S. flavescens regulated inflammatory response to intervene psoriasis (Biochem Pharmacol. 2013;85(8):1134-1144; Int J Clin Exp Pathol. 2018;11(11):5232-5240; Toxicol Appl Pharmacol. 2020;405:115209). However, the differences in efficacy and mechanism of these compounds in the psoriasis treatment, and the relationship between the chemical structures of them and their anti-psoriasis activity are still unclear, which is worthy of further study.
- The “KSCF regulated endogenous metabolites in 3 key pathways”, why further experiments haven’t been performed to verified these results?
Response: Thank you very much for the very good and novel idea from reviewer. The deep molecular mechanism of KSCF regulating endogenous metabolites in three key pathways is worthy to be further explored by detecting the key proteins in the pathways. Unfortunately, the biological samples including skin and serum in this study have been stored for too long ((about 18 months)), and the constituents in these samples may be significantly changed. Thus, these samples may not be used for further experiments, which will be meet in future research. This shortcoming has been supplemented in our manuscript (Lines 190-192).
- The “Institutional Review Board Statement” should be revised, as this study has involved animal experiments.
Response: As described by the reviewer, the “Institutional Review Board Statement” has been revised (Lines 334-337).
- Table 2 can be deleted, and describe the content in the main text.
Response: Thank you for your kind suggestions. Table 2 shows the details of the metabolic pathways, which thus is retained in the manuscript. However, the description of the table in the main text has been revised to show a significant difference (Lines 159-161).
- The abbreviation form “S. flavescens” can be used after its full name “Sophora flavescens Alt” has been given.
Response: We agree with what the reviewer described. The abbreviation form “S. flavescens” has been used in our manuscript (Lines 18, 20, 56, 58, 60, 188, 193).
- There should be a space between the numeral and unit.
Response: Thank you for your kind suggestions. The space between the numeral and unit has been checked and revised throughout the whole manuscript (Lines 235, 236, 240, 291, 300).

Reviewer 4 Report
Comments and Suggestions for Authors
Dear Authors,
I read and reviewed your Manuscript and I found it very interesting and significant. But there are a few points to work on:
1. It would be better for the understanding of methods used to put in the Introduction part a few references about interleukins and their role in the inflammatory responses.
2. In the Results and discussion section it would be good to add some results for other compounds investigated (if there are any) for fighting psoriasis and compare results with your research.
3. Figure 7 should be in the 2.5. section where the discussion about the metabolic pathways is.
4. In the References, plant species names and Genus names should be in italic.
Kind regards
Author Response
Comments
- It would be better for the understanding of methods used to put in the Introduction part a few references about interleukins and their role in the inflammatory responses.
Response: As described by the reviewer, the interleukins has been introduced in the introduction part (Lines 43-47).
- In the Results and discussion section it would be good to add some results for other compounds investigated (if there are any) for fighting psoriasis and compare results with your research.
Response: Thank you very much for the very good and novel idea from reviewer. It is reported that another flavonoid (kurarinone) and alkaloids (matrine and oxymarine) isolated from S. flavescens regulated inflammatory response to intervene psoriasis (Biochem Pharmacol. 2013;85(8):1134-1144; Int J Clin Exp Pathol. 2018;11(11):5232-5240; Toxicol Appl Pharmacol. 2020;405:115209). However, the differences in efficacy and mechanism of these compounds in the psoriasis treatment, and the relationship between the chemical structures of them and their anti-psoriasis activity are still unclear, which is worthy of further study. These have been supplemented in the Results and discussion section (Lines 187-190).
- Figure 7 should be in the 2.5. section where the discussion about the metabolic pathways is.
Response: We agree with what the reviewer described. The Figure 7 has been moved to the Results and discussion section (Line 187).
- In the References, plant species names and Genus names should be in italic.
Response: Thanks for your kind and professional suggestion. Plant species names and Genus names in the References has been revised to italics (Lines 363, 365, 368).

Round 2
Reviewer 3 Report
Comments and Suggestions for Authors
The comments have been well addressed, and the manuscript has been carefully revised.